

# CÆLIS: Software for assimilation, management and processing data of an atmospheric measurement network

David Fuertes [1,2], Carlos Toledano [1], Ramiro González [1], Alberto Berjón [1], Benjamín Torres [2], Victoria E. Cachorro [1], and Ángel M. de Frutos [1]

[1]Group of Atmospheric Optics, University of Valladolid (GOA-UVa), Spain
[2]GRASP-SAS, Lille, France

*Correspondence to:* David Fuertes (david@goa.uva.es)

**Abstract.** Given the importance of the atmospheric aerosol, the number of instruments and measurement networks which focus at its characterization are growing. Many challenges are derived from that: standardization of protocols, monitoring of the instrument status to evaluate the network data quality, manipulation of large volume of data and distribution of data (raw and processed). CÆLIS is a software system which aims at simplifying the management of a network monitoring the instruments,

processing the data in real time and offers to the scientific community new tools to work with the data. CÆLIS has been successfully applied to the management of the photometer calibration facility managed by the University of Valladolid in the frame of AERONET. The present work describes the system architecture of CÆLIS and some examples of applications and data processing.

## 1 Introduction

The atmospheric aerosols are defined as solid or liquid particles suspended in the atmosphere. Many studies have shown the importance of aerosols which play an important role in global climate balance and human activities. As direct impact the aerosol particles produce a cooling effect in the atmosphere, provide nutrients to the oceans, and directly affect human health. As indirect impact they change the chemical composition of clouds and therefore their radiative properties, lifetime and precipitation. To improve the knowledge of distribution and composition of aerosols is one of the emerging challenges

highlighted by the last IPCC report (IPCC, 2014) where it is shown that they have the largest uncertainty for the estimates and the interpretations of the Earth's changing energy budget.

Atmospheric aerosols can be monitored by remote sensing or in situ methodologies. Many instruments, such as satellites, ground-based instruments or instruments on board aircrafts, have been applied to the study of aerosols. Combination of instruments is also possible to exploit at maximum the synergies. For example: satellites have demonstrated the potential of high

spatial coverage and resolution, and standardized ground based networks the benefit of high accuracy. A common exercise is to validate satellite data with ground based networks (ground-truth).

One of these ground based networks is the Aerosol Robotic Network (AERONET, Holben et al., 1998). Leaded by NASA and PHOTONS (PHOtométrie pour le Traitement Opérationnel de Normalisation Satellitaire, http://loaphotons.univ-lille1.fr/)




AERONET is built as a federation of subnetworks with highly standardized procedures: instrument, calibration, processing and data distribution. It was created in the 90's with the objective of global monitoring of aerosol optical properties from the ground, as well as to validate satellite retrievals of aerosols. The standard instrument used by the network is the sunphotometer Cimel-318. It is an automatic filter radiometer with 2-axis robot and 9 channels covering the spectral range 340nm up to 1640nm. It collects direct sun and lunar measurements, and sky radiances in the almucantar, principal plane and hybrid configurations. Once the data are validated through instrument status and cloud-filtered, Aerosol Optical Depth can be obtained as direct product for the 9 wavelengths. Using inversion algorithms (Dubovik and King, 2000; Dubovik et al., 2006), many other parameters can be retrieved, such as size distribution, complex refractive index, portion of spherical particles and single scattering albedo.

The Group of Atmospheric Optics at Valladolid University (GOA) is devoted to the analysis of atmospheric components by optical methods, mainly using remote sensing techniques such as spectral radiometry, lidar, etc. One of the main tasks of the group is the management of an AERONET calibration facility since 2006, which is now part, together with the University of Lille and the Spanish Meteorological Agency, of the so-called AERONET-Europe central facility of the Aerosols, Clouds, and Trace gases Research Infrastructure (ACTRIS) research infrastructure. ACTRIS is since 2016 in the roadmap of the European Strategy Forum for Research Infrastructures (ESFRI). The GOA calibration facility is in charge of the calibration and site monitoring of about 50 AERONET sites in Europe, North Africa and Central America.

AERONET standards imply an annual instrument calibration and maintenance, as well as weekly checks to the observation data. The calibration process takes about 2 - 3 months and includes a post-field calibration for Sun, Moon and sky channels, maintenance of the instrumentation and a pre-field calibration for the next measurement period. In order to avoid gaps in the data sets during calibration periods, it is frequent that one instrument is swapped out with a fresh calibrated one. The network management implies to know where is each equipment, what is its exact configuration and calibration coefficients, and how many days remain until a next calibration is needed. During the regular deployment period the instrument has to be regularly checked to guarantee the data quality. A routine maintenance protocol is performed by the site manager but the network is the final responsible of data quality. The routine maintenance helps in reducing instrument failure and data errors but even with the best daily protocol, instrumentation problems may appear. Data monitoring made at the calibration center helps to early identify problems. However such a work cannot be accomplished manually in near real time (NRT) for a large number of sites.

In this context, the calibration facility at GOA had the necessity to implement some automatic mechanism (in addition to the standard mechanism of AERONET) to help the management of the network and facilitate the weekly data checks to guarantee the quality of the data. The origin of the CÆLIS system is to fulfill these two requirements. The system has to be designed to save all data, metadata and ancillary data (assimilated from other sources) in order to: On the one hand, support the management, maintenance and calibration of the network. And on the other hand, process the raw data in near real time (NRT) with different algorithms and provide to the network managers, site managers and ultimately to the scientific community a very powerful and modern tool to analyze the data provided by the observation sites. This work shows the fundamentals of CÆLIS system, both with respect to the scientific background and the information technology employed.





## 2 General architecture

CÆLIS has been designed to run on a server which, connected to internet, allows the external communication via a web interface. The software contains a *daemon* (in computing, a daemon is a background process that offers a service) which is the responsible to select and launch tasks. These tasks, later explained, take the responsibility to download new data whenever is

available and process it. Each task reads from the database the required input information and writes there the output. Some tasks use direct access to internet to retrieve data, e.g. download ancillary data from a FTP server. All information downloaded and treated by the CÆLIS tasks is stored in the database. This allows that following actions can retrieve all information required from the database (quicker extraction).

External users (organized by role with various privilege levels) can connect through the web interface to watch what tasks

are in execution and explore the results of finished tasks. All actions required by the system administrators can easily be done through the web interface. Network management is also performed through the web interface which allows, for example, to set up the installation of one instrument in a measurement station. The same information will be used by the system when data from the instruments reach the server and CÆLIS will compare the received information (instrument number, parameters, location, dates, etc.) with reference registers stored in the database (installation periods, configuration parameters, etc.) to know

if the instrument is working properly.

External systems, such as measurement stations, can be connected also to the server and submit data. Thanks to the web interface it can be done using the port 80 (standard HTTP) which avoid many problems derived from security rules of measurement stations (some of which are even in military areas).

The current system is managing 120 users and 80 stations. Each station can send thousands of aerosol observations every

year and the system is constantly growing. Some benchmark has been applied to know that current architecture can support a network 100 times bigger so that the database can grow safely in the future.

As shown in Figure 1, CÆLIS is composed by a database, a processing module and a web interface. These modules can be deployed independently even in different computers. The users and the stations interact with the system through the web interface. In the database, the raw data and metadata are stored, as well as retrieved products, ancillary data, user information,

etc. The NRT processing module is composed by the system daemon and a set of processing routines that extract information from the database, calculate products and write them in the database. The web interface is the platform designed to manage the system, manage the network and to provide visual access to the data and metadata, with tables, plots, searching capability, etc. Each of these elements will be explained in detail in the next sections.

## 3 Database model

Databases are one of the main concepts developed in the 80's in computer science field. Many different approaches have been developed with different success. There are many types of databases classified depending its characteristics. A database management system (DBMS) is a software with an interface to a database system that provides the user with advanced char-



acteristics such as the management of concurrency or a query language. The decision about what kind of database and which specific DBMS software is selected is one of the main design decisions since all developments will be impacted for it.

Relational databases are a traditional and well-known model, and have been successfully applied to many different fields. In relational databases the information is organized in tables or relations which represent entity types (Chen, 1976). A good database modeler is able to identify those entity types that are relevant, with the information that describes them. The tables or relations are composed by columns with the attributes that describe them, and rows which represent different individual entities that are identified by an unique key (one or more attributes that cannot be repeated in different rows). The tables are linked between them creating a relational model. The keystone of a database is a good design which needs to take into account the information that is targeted to be modeled as well as the way how the data is going to be accessed (to optimize the performance). Complex models with many groups of entities need to be planned in advance by creating an Entity-Relation diagram. This diagram then helps to the final implementation of the database which can be a direct translation of the diagram just taking some implementation decisions about a balance between data redundancy and performance.

The main elements of the entity-relation diagram of CÆLIS database are shown in Figure 2. The central entity is the Photometer, which produces Raw data. The photometer, with certain hardware configuration and calibration coefficients, is installed at one site of the network. The ancillary data for the site (e.g. meteorological data, ozone column, surface reflectance, etc.) need to be stored. Finally the measurement stations are supported by Institutions, which also can own instruments.

Each of these elements is in many cases representing a group of entities. For instance, calibration coefficients include extraterrestrial signal for the different Solar spectral channels, radiance calibration coefficients for Sky channels, coefficients for temperature correction of the signals, instrument field of view, etc. Another example is the hardware, that includes the different parts (sensor head, robot, collimator, control box, ...), the spectral filters with the corresponding filter response, etc.

The lower part of the diagram is closely related to the network management, with an inventory of all hardware parts identified with their serial numbers and related to the institution that owns them. The upper part is related to the raw data production and its organization is optimized for data extraction (to create products) and is consistent with the physical meaning and relevance of the quantities. The installations are manually introduced by the network managers, so that any data file submitted to the system from a measurement station, can be validated.

Other tables contain ancillary information that is needed to process data, such as the list of stations (including coordinates, etc.), global climatologies for certain atmospheric components (ozone, nitrogen dioxide, ...), Sun and Moon extraterrestrial irradiance spectra, or spectral absorption coefficients for several species (ozone, NO2, water vapor, etc.).

Many different DBMS can be used to implement such a model: OracleDB, SQLite, PostgerSQL, etc. CÆLIS is based on a MySQL database. The benefit of MySQL is that it is a software widely used by many different communities so, as result, the software is very robust, complete, stable, well documented and can be run over many different architectures.

The entity-relation diagram for CÆLIS shown in Figure 2 shows the fundamental part of the database, called level 0. On top of that, direct products (obtained with the combination of raw data, calibration coefficients and ancillary data) are stored. This represents 'level 1' products, and are physical quantities with their corresponding units and estimated uncertainties (derived from the calibration uncertainties). In our case, these products are basically aerosol optical depth, water vapor content, sky





radiances and degree of linear polarization of the sky light. On top of level 1, there are more sophisticated products, like those derived from inversion algorithms, as well as any flags or 'alarms' that are produced to help in NRT data quality control. Level 2 products use and combine previous level quantities to retrieve other parameters, but no longer go down to the raw data. For instance, the inversion codes by Dubovik (Dubovik and King, 2000) and Nakajima (Nakajima et al., 1996), use spectral

aerosol optical depth and sky radiances, to retrieve aerosol particle size distribution, refractive indices, single scattering albedo, etc. More advanced products, that combine photometer data with other aerosol data (e.g. lidar) also belong to this group, that is denoted 'level 2' products. A clear example is the GRASP algorithm (Dubovik et al., 2014, http://www.grasp-open.com/), that is able to digest data from different sensors (satellite and ground-based, active or passive) to provide a wide set of aerosol and surface parameters. The system architecture as described here is shown in Figure 3.

## 10  4  Processing chain and near-real-time module

As it is explained in the previous section, CÆLIS system provides many different data products. To provide each product, some input data has to be processed in a specific way. This is what we call "task". The job is divided in a set of simple tasks. The system works as a state machine: one task cannot start until the previous one is finished, no matter if the second task is dependent or independent from the previous one. When many tasks work sequentially to achieve a common objective, we

create a chain of tasks. The daemon running in the server is responsible to coordinate the different tasks, as it will be explained in detail in the next section.

The main processing chain in CÆLIS is the set of the tasks that are performed once the new photometer data are loaded in the system, shown in Figure 4.

The pre-filter checks that the file uploaded to the server is really a valid data file pertaining to the AERONET instruments

managed by CÆLIS. If this is true, the "filter" checks that basic information (instrument number, coordinates and dates, configuration parameters) is in accordance with the stored information about instrument installations. If any of these parameters does not correspond to its expected value, the data file is put to quarantine and the network managers receive an email notification ("send notification"). If all parameters are correct, the measurements are inserted in the database and the data file is forwarded to any desired destination: our data file repository, an AERONET server at NASA, etc. With the raw measurements inserted

in the database, the processing of level 1 products is triggered: the aerosol optical depth, water vapor, and radiances in several geometries (almucantar, principal plane, hybrid, cross). With all raw data and level 1 products, a set of flags concerning data quality control are produced by the "alarms" task. These flags are produced in near real time, as soon as new data are submitted to CÆLIS from a particular site. Since the automated QC analysis needs all available information, the "alarms" task is always the last one of the processing chain. The QC flags in near real time are a very important element in the network management.

More details are given in section 7.2.

Any new implementation, for instance a new level 1 product, needs to be inserted in the processing chain taking into account its dependency on any other elements in the chain. The last step in a certain task is to trigger the next one in the chain.





Whenever new data are found (photometer, ancillary, other) or new information is inserted by the managers (new calibration, installation, etc.), a processing chain is triggered. The management of all chains in CÆLIS is carried out by the daemon, which is explained in detail in the next section. This kind of task organization is highly modular, so new elements in CÆLIS, either data or different instruments, etc. can be added by creating new chains, that can be connected or not to the existing ones.

5 The near real time processing module (see Figure 1) is composed by a set of programs and libraries that are related to all the above mentioned tasks. These are programmed mainly in C for fast computation with large datasets and inter-operability with other technologies, allowing the possibility to use other languages in the future. A GIT repository is used to facilitate version control and deployment of the software.

## 5 Daemon

10 The daemon is the responsible to organize the tasks and to decide which process has to be run in each moment. It has to address different challenges:

1. Run scheduled tasks according to their priority.

2. Know which task must trigger other tasks.

3. Keep the sequence.

15 4. Optimize the server processing capability, running less important tasks when the CPU is idle.

The tasks are stacked in the system. Figure 5 is a representation of the stack of the tasks where the green tasks are actions that can be executed right now and the red tasks are disabled until the "date of activation" arrives. Each task is described by the following information: 1) Group: Classification of the task 2) Name: Name of the task. 3) Object: In case it exists, it defines the target where the task will be applied (for example, one file, one particular instrument, one site, etc.) 4) Date range: In case 20 it is defined, the task can be applied to a specific date range. 5) Date of activation: This mark allows to organize when the task can be run. Note that some tasks can be defined to be executed in the future. 6) Priority: It defines the importance of the task. Frequently, several tasks can be activated at the same moment. In these cases, this mark indicates the system the order in which the tasks should be run. At the same time, the processing chains (commented in figure 4) used this mark to indicate daemon the order of the tasks. When a task is executed, if it is part of a chain, it will introduce in the stack the next actions (sorted by 25 priority). This is the procedure to keep the system alive and always working.

In every moment, the stack contains the current tasks that can be executed right now, as well as, the tasks that are scheduled to be run in the future. This is the method used by the system to repeat tasks: if a task A wants to be repeated every 15 min, once it is executed the system will remove it from the stack but will add it again with the "date of activation" 15 min later.

After a task is executed, the information of the execution is saved into a log. This allows the system administrator to study 30 the behaviour of the system, to know what has been executed, to foresee the use of the system and to tune up the parameters of CÆLIS to keep a good balance between NRT actions and the load of the system. Figure 6 shows the log of actions and the



statistics of them. Thanks to it the system administrators can know how much time a specific task takes every day and how many times they are executed. This information is crucial for system administrators and developers: They can know which tasks take more time and why (in the cases when a defined task is too slow or is called many times, etc.) and create good plans to optimize the system.

In a regular situation, the system works automatically. For instance, when the daemon starts, the common operations are introduced in the stack of task. One of these common operations will look for new data and metadata with a certain frequency (e.g. once per hour). Then, if the task in charge of looking for new data find them, it will add new tasks to process those data, triggering the processing chain. In the case that there are no new data, the task will add the same task to find new data some minutes later.

The system administrators can add tasks manually and they can change the priority of the current tasks in the stack. One of the main manual tasks that the administrator can add is the "stop" action. The "stop" task has a duration of a few seconds and, once it finishes, it re-enters itself in the stack. This process continues until the administrator erase the task manually from the stack. Depending on the priority assigned to the "stop" task, the system can be completely blocked or it can do some tasks with high priority. Thus, if the "stop" task is introduced with the highest priority it will be the only task executed. Another main

action is to shut down the system: If the system administrator wants to shut down the daemon, this task should be introduced. This guarantees that the system is turned off when is idle and no task is interrupted sharply.

The system is also prepared for a sudden shutdown (for example, power outage). Given that the system only removes the tasks from the stack when they are finished, once the server is turned on, the first task to be executed will be the one that could not be completed. The fact that all these scenarios are taken into account by the stack of tasks, make CÆLIS to be a robust

system and easy to maintain.

Regularly, the system executes maintenance tasks. For example, a daily backup is performed. This task is scheduled every night thanks to "date of activation" information. The maintenance tasks can cover many different activities that need to be done regularly into the system. Other examples of the maintenance tasks are the optimization of the database, regular rebooting, etc.

The current implementation of the daemon is developed using bash scripts. This characteristic allows to run tasks written in

any language. It is planned to improve the current implementation by using Python language and to introduce parallelism into the tasks chain. If it has not been done until now is due to the relative low load of the system and to process in sequential mode is enough to provide data in NRT. When more sophisticated algorithms will be run (such as inversion retrieval algorithms) a new implementation of the daemon will be needed. Alternatively, tasks can be launched in a server farm allowing the system just to organize the tasks, keeping its load really low. The tasks are currently implemented mainly in C because of its high

performance but any compilable language is allowed in the server.

## 6   Web tool

CÆLIS system offers a web interface to allow the users interact with it: www.caelis.uva.es. The web interface is a high level view of the data model thus it will show the information in real time, as soon as it is processed. The web system is secured with



a private access only for registered users. During the registration process a "role" is assigned to each user. The roles allows to identify groups of users with different permissions into the system, for example, regular users (site managers or researchers) that can access to its data or a sysadmin that can handle the stack of tasks of the system or rebooting it.

The web interface acts as high level access to the database. It can extract and relate different data and show them all together as an online real-time report. CÆLIS has implemented many different use cases that fulfill all common actions performed by the users. The system offers different tools depending on the role of the user. For example a site manager can check the performance of one instrument, a network manager can modify the location of an instrument or a system administrator can check the tasks that the NRT module is executing. The web interface allows the user to explore the database showing tables and customizable plots. Some of these use cases are described below in the example section. The web interface allows to query information as well as inserting new information. This is specially interesting because it constitutes the second way to insert data in the system: data inserted by the users (data can also be registered by the tasks controlled by the daemon, see previous section). In the case of users, since they work via web interface, the actions can be controlled in two senses: 1) The user has the permission to introduce the information; and 2) the information introduced is validated. Also, the fact that manual information is registered by means of the web interface allows the system to launch "derived tasks" for an specific action. For example, in case that a new measurement site is created the system can add to the stack of tasks the action of "insert climatological data for the new coordinates". Everything is triggered automatically and controlled by the logic implemented in the web interface.

The web interface has been developed using PHP through the Symfony Framework, Bootstrap as CSS framework, jQuery as javascript framework (used for asynchronous actions and to dynamize the interface). The choice for PDO (PHP Data Object) is Propel. Every technology selected in the development has been highly studied:

– PHP is a widely extended language for web developments. It shows very good performance and huge websites has been developed using it. It has a big community behind it, which offers helpful support. The ecosystem (libraries implemented to be used with PHP) is one of the best for web computing: there exist libraries for graphical representation, maths, etc.

– Symfony framework: nowadays for a quick development the use of frameworks is heavily extended. They allow the developers to stay focused on the main issues and get rid of the complexity of common things: user management, database access, etc. The selection of Symfony instead of other alternatives was done because it was getting a lot of popularity at the moment of starting the development of the web interface and the community was very active. Also, it is easy to use, contains hundreds of libraries, it is powerful, flexible and it shows very good performance.

– Javascript is used for asynchronous connections with the server in order to offer very dynamic interface. Javascript is the undisputed winner for this purpose.

– JQuery is used as Javascript framework. There are alternatives but JQuery is very well integrated with Symfony, it is popular and fulfills all system requirements.





- Propel was selected as library for PDO because it allows to have primary keys with multiple fields. CÆLIS database has been deeply optimized and the use of multi-field primary keys improves a lot the performance in comparison with an auto increment id (common alternative). At the moment of the choice only Propel managed such kind of keys.

The web interface has been structured and designed to grow, managing other measurement networks or scientific projects. Those projects can share the core of the developed code. This allows to start projects from existing code instead of starting from scratch, making it really quick to add new features. For example, one of these common utilities is the plotting tool. CÆLIS has a really powerful and flexible tool to plot the data. The tool is implemented in the server side using PHP. The benefit of this approach is that, in case of plotting really huge pieces of data, the plotting is still quick since the data is not transferred to the client (the web browser). Instead, the plot is created in the server and only some tens of KB are transferred to the users. This solution is optimal for treating large pieces of information. A disadvantage of this approach is that the result is less dynamic than an implementation on the client side.

The web interface also implements web services for the machine to machine communication. These web services allow to perform common operations such as data transfer from the measurements sites to the server. A great advantage of this approach is that even well secured external machines can connect to a HTTP server. Sometimes, as in military bases, they need special permission to set up internal proxies and allow access to the system but it is widely accepted that HTTP protocol on port 80 can be used everywhere. Alternatively CÆLIS can offer other data transfer alternatives like FTP or via e-mail, but the most common is to use the web services.

## 7 Examples

In order to illustrate the capabilities of the system we will now show a set of examples, focusing on the typical needs of the different users: site managers, calibration center (network managers) and researchers.

### 7.1 Site manager use case

Site managers are interested in knowing the status of their instruments and general information about the instruments and their sites. This example will illustrate how a site manager can access to all this information about, checking what is the metadata related to its instrument.

CÆLIS offers access to all metadata related to each instrument: calibration coefficients, temperature corrections, configuration parameters, filters, etc. The metadata are in general different for each deployment period.

Figure 7 shows the description of the sunphotometer #783 of the AERONET network (registered in CÆLIS and calibrated by GOA). There are three blocks of information: 1) metadata, such as calibration coefficients or configuration; 2) network management information such as, deployment periods (sites, dates); 3) administrative information, such as hardware inventory of all parts of the instrument.

The continuity of the datasets is an important issue in AERONET. In order to avoid (or minimize) data gaps, it occurs often that a calibrated instrument is sent to a site to swap out an instrument that needs to be calibrated. Hence a number of





instruments is rotating inside the network, from site to site. This fact makes it difficult to monitor which instrument is where. CÆLIS offers the information about each site, with the list of instruments and deployment dates in that particular site. This is all easily accessible to site managers.

The illustration in Figure 8 shows the information of the measurement site placed in Madrid. Some general information is
shown on top of the page, followed by the list of instruments and measurement periods. This information is linked with the instrument information showed in the previous figure in a way that is really easy to dive in the information.

### 7.2   Network manager use case

One of the most important applications of CÆLIS is the real-time data monitoring. This information is used by the network managers (as well as site managers) and it provides useful information about the instrument performance. The biggest benefit
generated for this powerful tool is that it allows identifying problems in the instrumentation as soon as they appear, rising a flag automatically. The network managers at the calibration center can send a warning to the site managers. If the site managers resolve the problems quickly this generates an improvement in the data quality thus, in an improvement of network quality. The calibration center is continuously monitoring this information in order to warn and assist the site managers in case that a problem is not quickly solved.

When the system receives new data file from a measurement site, it processes the data generating new products. From raw measurements it calculates the aerosol optical depth, sky radiances and other products. The last product in the processing chain triggered by the arrival of new data is the 'alarms'. This product studies the new data, metadata and derived products in order to identify malfunctions in the instrumentation.

Figure 9 shows an screenshot of CÆLIS web interface where we can see the status of a specific site in the last 6 days (it is
a calibration site so it has more than a single instrument). This view is highly configurable thanks to filters (sites, instruments, dates, etc. ). Finally, useful information can be triggered just by clicking over specific places. For example, when the photometer number is clicked, instrument status information is shown as graphs (battery voltages, internal temperature, etc.); or when a specific day is selected the user can explore all information (raw data, products) received and processed for that particular day.

Calibration centers need to solve every day instrument issues and multiple questions and this is only possible thanks to a real
and deep knowledge of the instrumentation. CÆLIS helps in the routine problems and provides very useful information about the data contained in the database. As a quick way to explore the database with the freedom that an unknown issue imposes, the system offers to the users a data viewer which allows customizing the data that the user wants to display.

The figure 10 shows how setting up a specific case in which data from different sources are shown in the same plot in order to help the network manager to understand the problem. We can select one or multiple variables (all available raw data and
data products) from one or multiple instruments, and display them for a particular date-time range, with full flexibility in plot configuration (colors, axis, etc). Specifically, the example shows battery voltage and robot errors. The plot clearly indicates that the power supply stopped working, therefore the battery is loosing charge and the robot cannot operate normally and returns robot errors in coincidence with the battery voltage decreasing trend.



## 7.3 Research applications

Up to now we have shown the capabilities of the system with focus on giving solutions to routinary tasks related to the network. But we cannot forget that the system was born in a research group whose main objective is to study the atmospheric aerosols. CÆLIS also offers good tools to achieve that objective. As it was mentioned during the explanation of the web tool, CÆLIS

was created so that it can grow and include new projects. The following example will show a specific project of the research group with the aim of studying the AERONET database. It creates an organized database from AERONET data and allows the users to answer general questions about AERONET sites, such as: since when an AERONET site is active; how much data (and in which quality level) a site has, etc. This project re-uses at maximum the core of the system (users access, plot tools,...) and lets the developers to create a new tool quickly. CÆLIS has been used as 'framework' for data analysis. The effort to develop

this system is far less than starting from scratch. The features needed in this development have been a tool to assimilate the new data and the specific views that show the results. Additionally, CÆLIS can re-use the database added here in other projects.

Figure 11 shows the automated aerosol data analysis of an AERONET site (in this case, Palencia site). In it, we can see the data coverage (for level 1.0, 1.5 and 2.0 of the AERONET database), monthly statistics of aerosol optical depth and Angstrom exponent, frequency histograms and AODvs.AE scatter plot providing basic aerosol type classification. These plots provide a

general overview of the site characteristics in terms of data coverage, aerosol statistics and type classification, which can be used as a first approach in order to select a site for some particular study. Then, based on this general information, we can ask other questions that we can resolve by interrogating the database directly. In order to illustrate how it can be done, next part of the example will show how to identify special aerosol events at the Palencia AERONET station. For this purpose, we need to explore the CÆLIS database. The starting point will be the following questions: How many days of high turbidity do occur in

the site? And, how many of them can be classified as desert dust? To answer these questions we will use the previous (overall) climatology and we will do some assumptions. First, we are going to assume that 'a high turbidity event' takes place when the AOD average is larger than the climatological mean plus 2 standard deviations. Based on that assertion, we can write a SQL query that is launched in the system database and will review every single value and return the result. To create this SQL query it is important to have accurate knowledge of the database model in order to obtain the expected results and within a

reasonable time. For our particular case, we will use the table cml-aod which contains the general information about each AOD measurement, and cml-aod-channel which contains the AOD information about each specific filter (wavelength channel). First, we are going to check how many days with AOD measurements we have for Palencia AERONET site:

> SELECT COUNT(a.'date') FROM 'cml-aod' a WHERE station='Palencia' GROUP BY DATE(a.'date')
> Result = 2730

Now, we will filter this result by checking how many days the AOD 440nm has at least 10 observation points greater than

0.31 (climatological AVG= 0.13 and STD=0.09):



> SELECT DATE(a.'date') FROM 'cml-aod' a LEFT JOIN 'cml-aod-channel' c
> ON a.'ph'=c.'ph' AND a.'date'=c.'date' WHERE station='Palencia'
> and c.'wln'=0.44 and c.'aod'> 0.31 and cloud-screening-v2='cloud-free'
> GROUP BY DATE(a.'date') HAVING COUNT(*)>10 ORDER BY DATE(a.'date')
> Result 285 days: 2004-02-12 2004-02-14 ... 2017-03-14

Finally, we will make another assumption: A desert dust event must present a low value of the Angstrom exponent, lower than the average minus 2 times the standard deviation (climatological AVG=1.29, STD=0.37)

> SELECT DATE(a.'date') FROM 'cml-aod' a LEFT JOIN 'cml-aod-channel' c
> ON a.'ph'=c.'ph' AND a.'date'=c.'date' WHERE station='Palencia'
> and c.'wln'=0.44 and c.'aod'> 0.31 and a.'alpha-440-870'<0.55
> GROUP BY DATE(a.'date') HAVING COUNT(*)>10 ORDER BY DATE(a.'date')
> Result = 65

These are very strong conditions, that identify the most intense dust event days over the site. Finally, we will show for one selected year (2016), the number of dust event days per month identified by our assumptions:

> SELECT MONTH( 'date' ) , COUNT( * ) FROM ( SELECT DATE( a.'date' ) AS 'date'
> FROM 'cml-aod' a LEFT JOIN 'cml-aod-channel' c ON a.'ph' = c.'ph'
> AND a.'date' = c.'date' WHERE station = 'Palencia' AND YEAR( a.'date' ) =2016
> AND c.'wln' = 0.44 AND c.'aod' > 0.31 AND a.'alpha-440-870' < 0.55
> AND cloud-screening-v2='cloud-free'
> GROUP BY DATE( a.'date' ) HAVING COUNT( * ) >10
> ORDER BY DATE( 'date' ) ) dd GROUP BY month( 'date' )

The result is shown in Figure 12, where we can see the two peaks of occurrence of Saharan dust episodes over Spain, i.e., February-March (early spring) and May through September, basically the summer months.

This example shows the flexibility and power of a relational database to make data analysis. Through SQL queries very complex customized questions can be asked and the data can be easily extracted from the database.

## 8   Summary and Conclusions

The atmospheric aerosol particles are one of the most important contributors to the climate forcing uncertainty. They are nowadays extensively measured from ground and space, with very different techniques. It is therefore important to develop tools using modern technologies to monitor (quality control), process, analyze and combine those data.

This paper has described the CÆLIS software tool, which has been developed for the management of the photometers that are calibrated and monitored by the calibration facility at University of Valladolid, as part of AERONET. This is a NASA program devoted to the measurement of the atmospheric aerosols and the validation of satellite-derived aerosol products. CÆLIS is



intended to provide management of the photometer network, archive the data and allow data analysis and research. The use of this kind of advanced system reduced the number of human errors, because well-tested software performs the routine tasks in an automated way.

The core of the CÆLIS system is built in a relational database. It stores user information (with its privileges), data, meta-
data, etc. Around this database, different modules use it and offer different services: a web interface to explore the database and a near-real-time (NRT) module to perform different processing. All this software can be re-used for extending the system, for instance with other instrument types, etc.

The construction of the database requires a balance between normalization and redundancy. The current system has three different levels of data. Level 0 contains the raw data and the network management information; level 1 contains direct products;
and level 2 contains advanced derived products that can be calculated. Each level is based on the information of the previous one. A keystone of the system is to have correct model of the first level, i.e. normalized and without redundancy. This helps to keep the congruence of the system. Based on these data, other products can be developed. Depending on the use of these products, some redundancy can be necessary. For instance, pre-calculated products can allow fast visualization in the web interface, that would be too slow to be done on-the-fly.

The existence of redundancy implies that automated tasks are needed to keep the congruence. This is done by the NRT module, which organizes the actions in separated tasks. The NRT module is always alive thanks to a daemon which is based on a stack of tasks organized by priority, and that decides in every moment what it needs to be done.

Users (site managers, calibration centers, researchers, ...) can use the web interface for quick access and visualization of data. The relational database is shown to be an appropriate tool for research because it allows performing queries and extracting data
in a fast and very flexible way.

*Acknowledgements.* The authors gratefully acknowledge the effort of NASA to maintain the AERONET program. This research has received funding from the European Union's Horizon 2020 Research and Innovation Programme under grant agreement No 654109 (ACTRIS-2). The funding by MINECO (CTM2015-66742-R) and Junta de Castilla y León (VA100U14) is also acknowledged. We thank all the users of CÆLIS for their feedback, especially E. Cuevas, C. Guirado and R. Román.



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

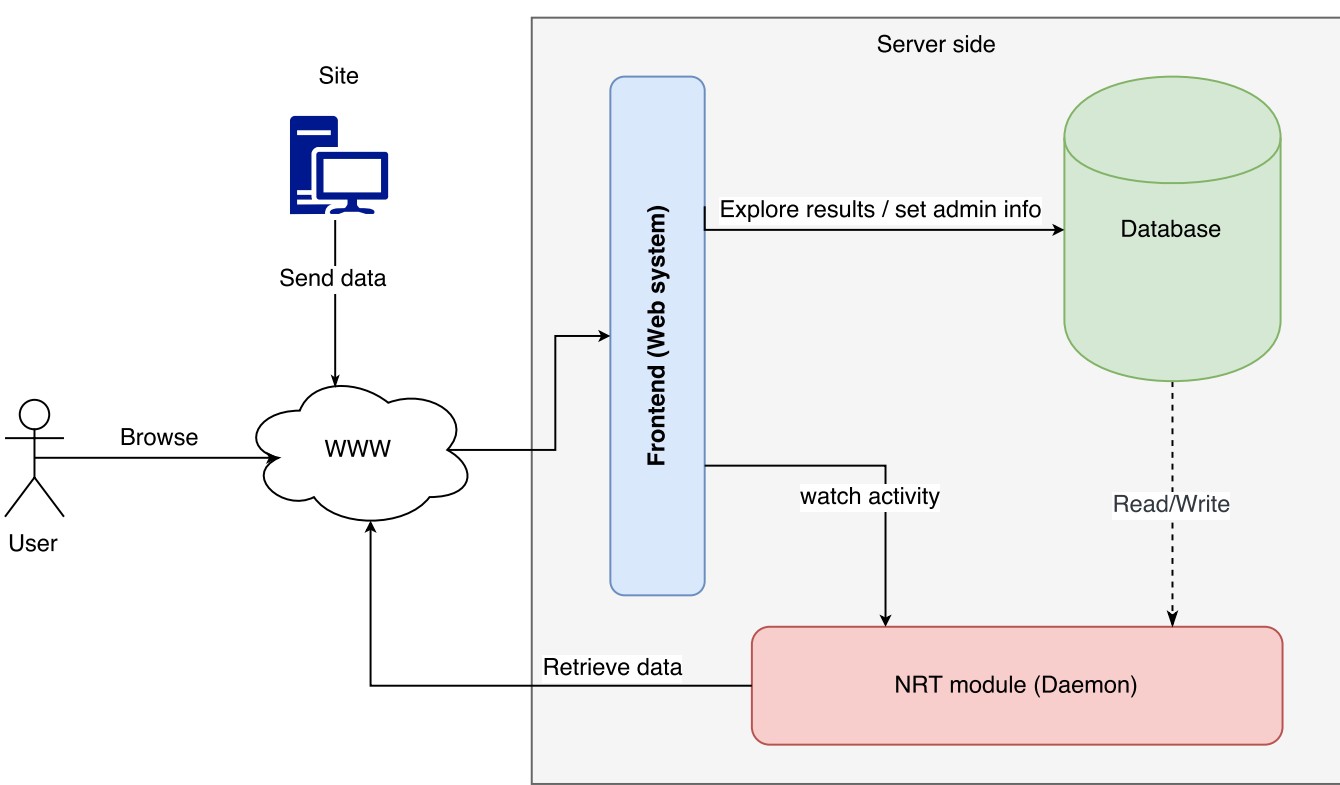

**Figure 1.** Diagram of CÆLIS architecture. Arrow directions indicate who initiates the action (data flow is always bi-directional)





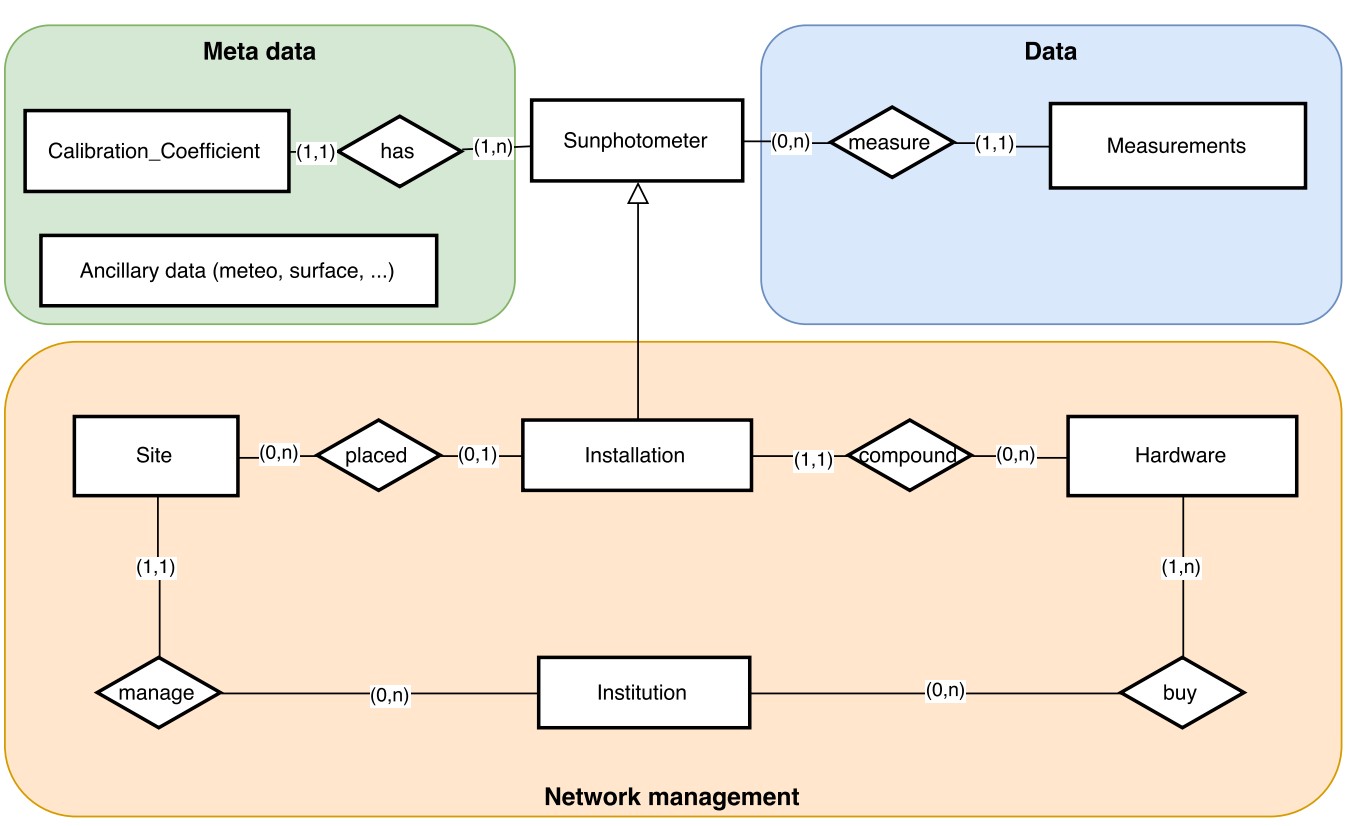

**Figure 2.** Entity-relation diagram (Chen, 1976) for CÆLIS (extract of the main elements). Entities has been divided in three logic blocks.

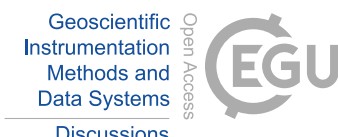



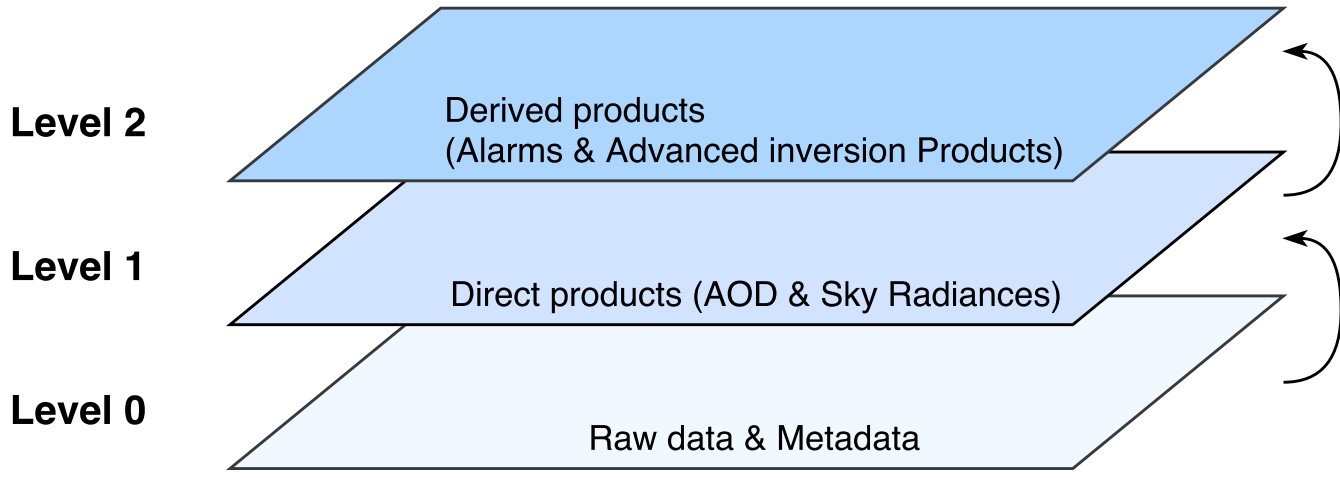

**Figure 3.** Different logic data levels. Each level is based on the information of previous level.



**Figure 4.** Different processing chains and their relations. An action triggers a task and then the actions bubble up.



| Acc | Sub acc | Object | First | Last | Valid | PRTY |
|------|-----------|-----------------------|---------------------|---------------------|---------------------|------|
| cimel | sendaeronet | 20170510_p382_1218.K7 | | | 2017-05-10 13:59:26 | 4420 |
| cimel | processaod | 382 | 2017-05-10 08:23:27 | 2017-05-10 12:13:09 | 2017-05-10 13:59:26 | 4410 |
| caelis | sitecheck | | | | 2017-05-10 13:59:26 | 3900 |
| cimel | finddata | | | | 2017-05-10 14:12:03 | 4400 |
| gdas | finddata | | | | 2017-05-10 17:02:43 | 4050 |

**Figure 5.** Example of CÆLIS tasks stack. Order is related to next task to be executed. Green are task that can be already selected to be executed by contrast of red tasks, tasks to be executed in the future (when valid time arrived). The example is captured at 14h00 on 2017-05-13



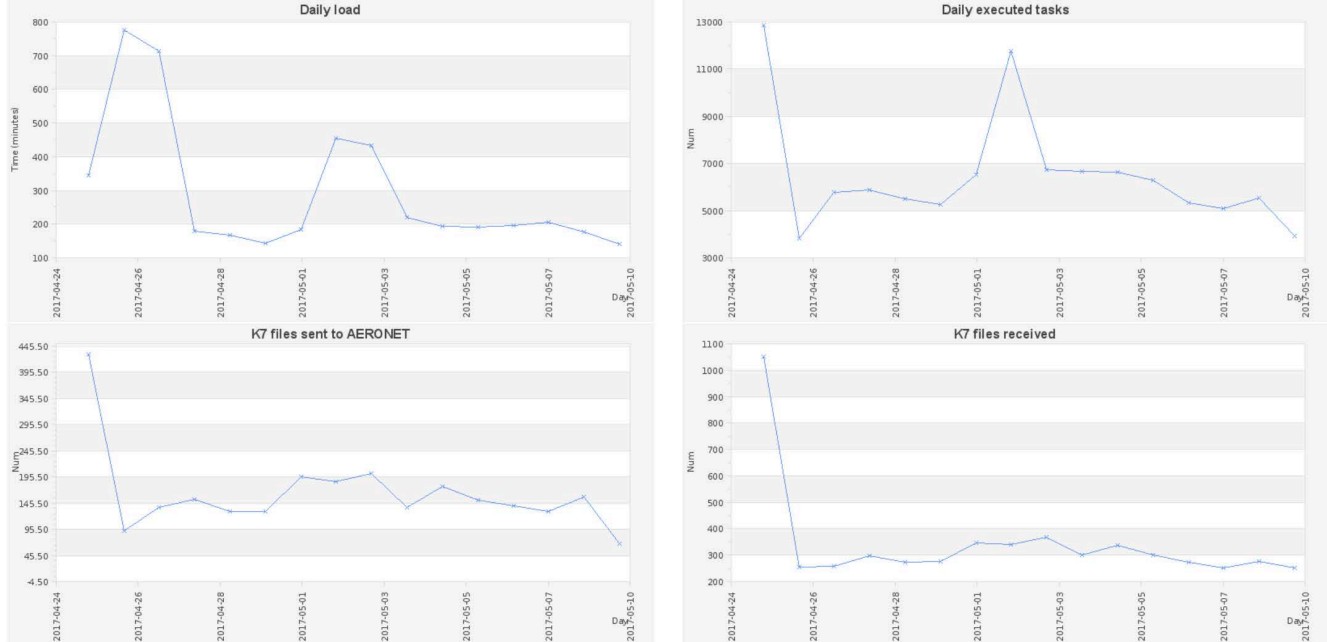

| Acc | Sub acc | Object | First | Last | Valid | Time to Start | PRTY | Start | Duration | Status |
|------|----------|---------|--------|-------|--------|------|------|--------|----------|--------|
| cimel | processaod | 243 | 2017-05-09 19:03:44 | 2017-05-10 10:17:56 | 2017-05-10 11:08:22 | 25 seg | 4410 | 2017-05-10 11:08:47 | 0.5 seg | OK |
| cimel | sendaeronet | 20170510_p243 _1008.K7 | | | 2017-05-10 11:08:22 | 23 seg | 4420 | 2017-05-10 11:08:45 | 2.3 seg | OK |
| caelis | sitecheck | | | | 2017-05-10 11:03:26 | 3 min | 3900 | 2017-05-10 11:06:33 | 1.5 min | OK |
| ... | | | | | | | | | | |

**Figure 6.** Plots represent CÆLIS load from different point of views: minutes of CPU per day, number of tasks executed per day, sunphotometer raw data files sent to AERONET and received from stations. Those plots are constructed based in log information. Table below shows an example of the log.





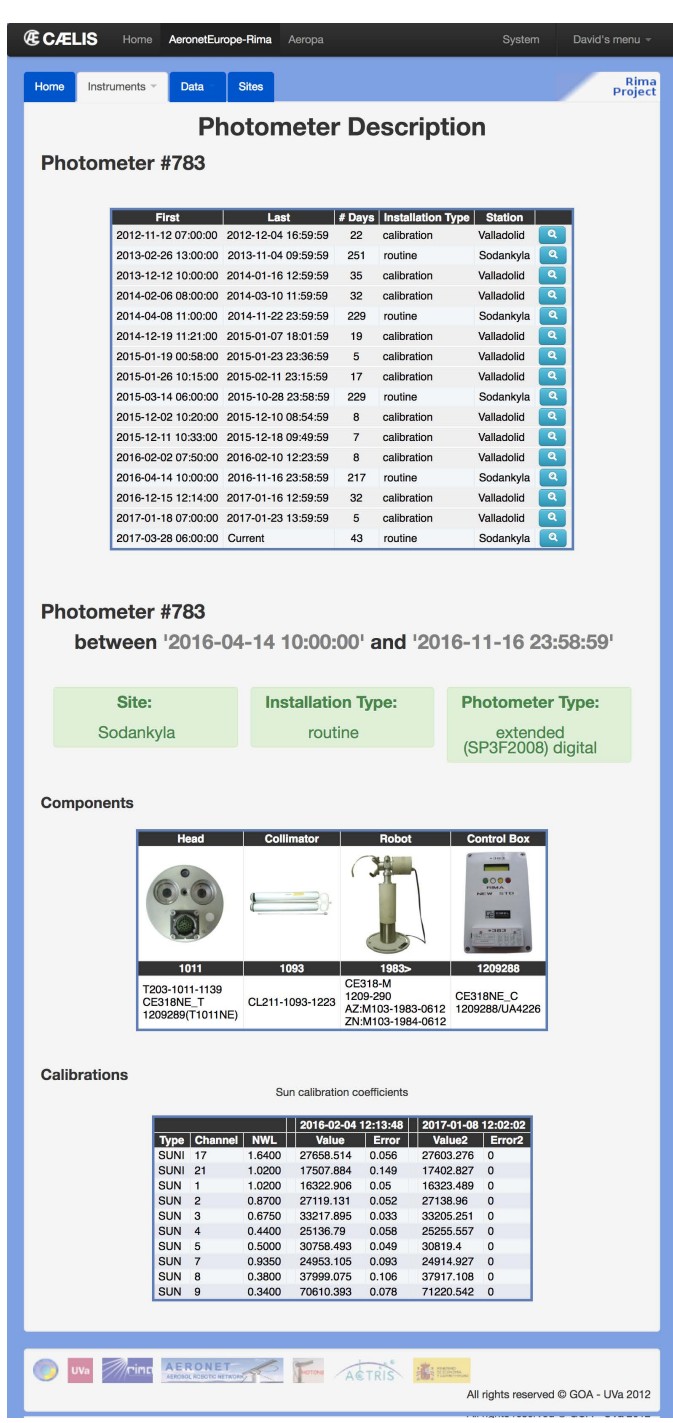

**Figure 7.** Screenshot (cutted and abbreviated) of sunphotometer description for AERONET instrument #783.





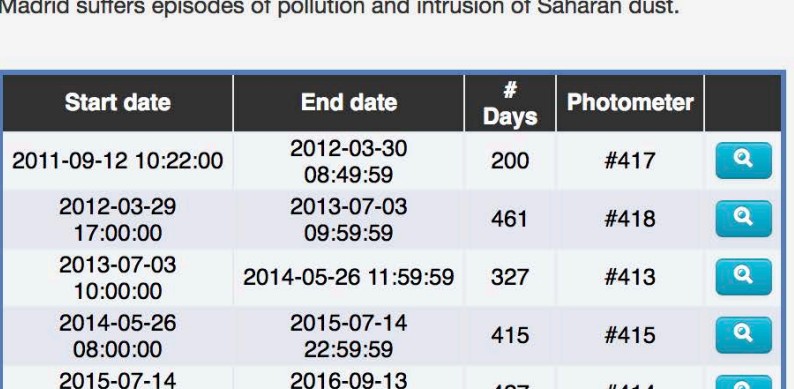

Figure 8. Screenshot of the Madrid 'site description'





**Figure 9.** View of alarms for a specific day at Valladolid site. A zoom shows how is the signal of a good day and how a problem is automatically identified. Specifically the sunphotometer #618 on 2017-04-10 has almucantars where the sun is not in the center (it is usually a head cable too tight).





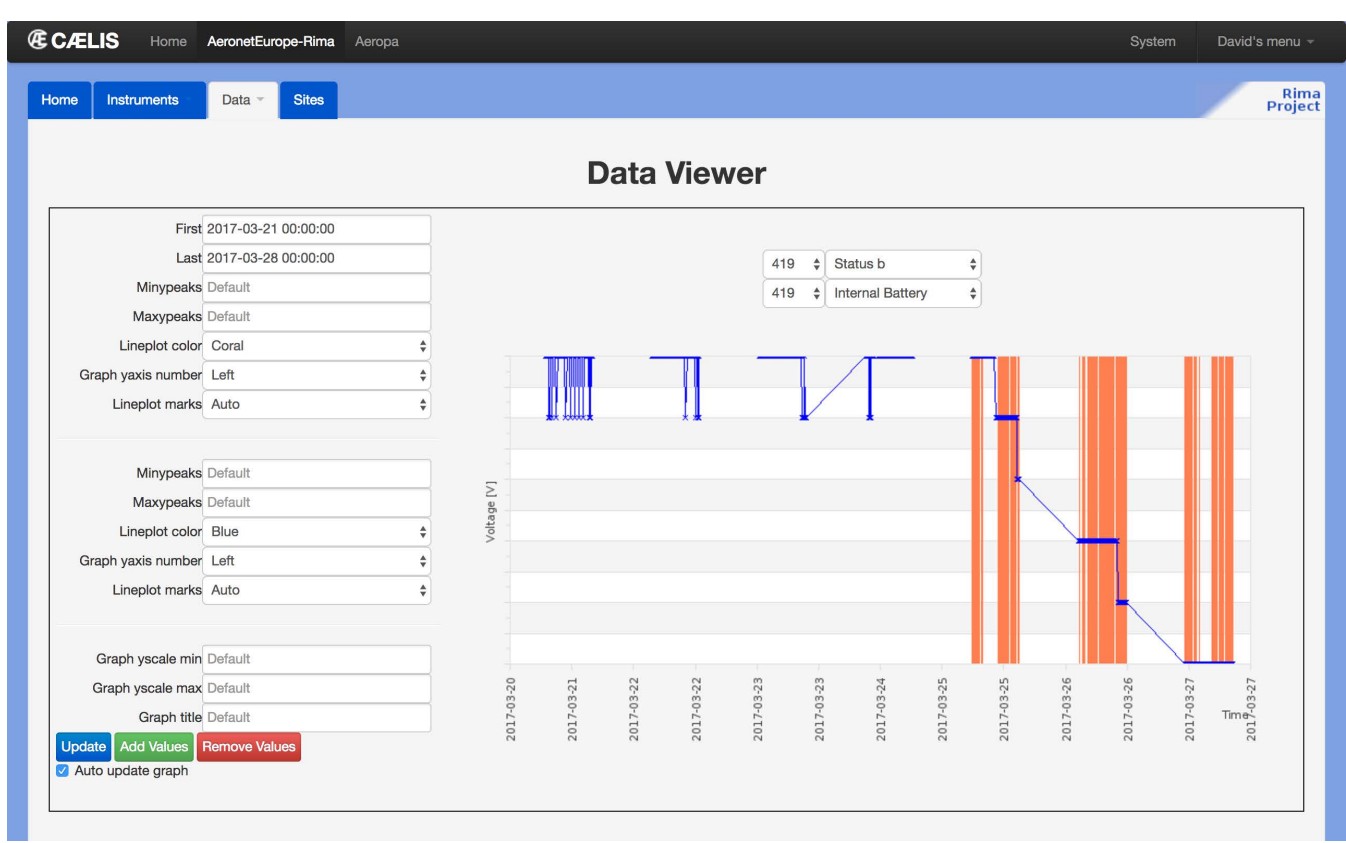

**Figure 10.** An example of data viewer where an instrumental error can be identified. The power supply is disconnected and the internal battery decrease drastically. When energy it is not enough to move the robot, status errors appear (each orange vertical line represent an status error in an specific time).



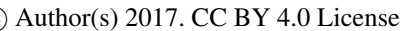

**Figure 11.** Statistical analysis of aerosol optical depth (AOD) and Angstrom exponent (AE) derived from AERONET for Palencia site in 2016: (a) data coverage for level 1.0, 1.5 and 2.0 of the AERONET database; (b) Aerosol type classification based on AODvs.AE scatter plot. (c) AOD (440nm) monthly statistics using whisker box; (d) Frequency histogram for AOD (440nm); (e) AE monthly statistics using whisker box; (f) Frequency histogram for AE.





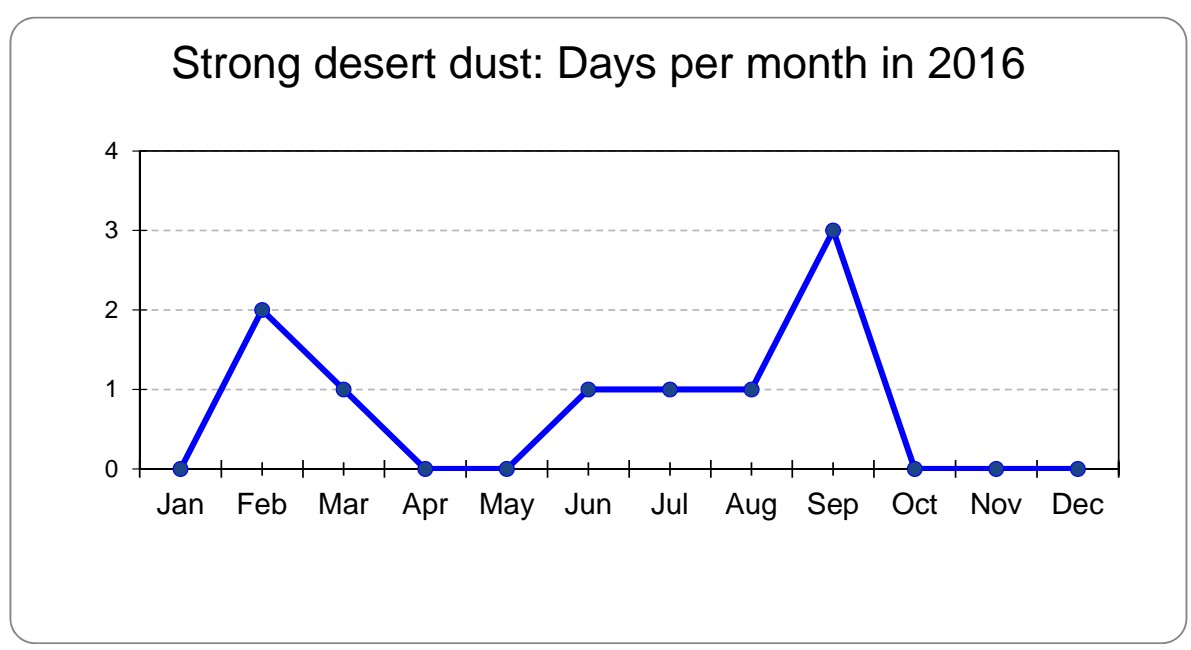

**Figure 12.** Number of strong Saharan dust event days for each month of the year 2016 over the 'Palencia AERONET site (Spain) derived from a query to the CÆLIS relational database (see text).