# Peer review of "CÆLIS: Software for assimilation, management and processing data of an atmospheric measurement network"

_Geoscientific Instrumentation, Methods and Data Systems, 2017_

## Referee Comment (RC1) · Anonymous Referee #2 · 6 Sep 2017

1. General Comments The authors present and describe new software (Caelis) developed to assimilate, manage and process aerosols-related data recorded by the Aerosol Robotic Network. Although the description and performance of Caelis is explained with sufficient detail, this reviewer thinks that that the manuscript could and should greatly benefit from showing how specifically Caelis improves its predecessor software. Also, the use of English language must be substantially improved. This reviewer's recommendation is that the paper may be accepted for publication after both aspects and few minor issues listed below are addressed.

2. Specific Comments

[Figure]

Page 1, L13: It is widely agreed that the overall effect of aerosols is to cool the climate. However, aerosols can also warm up the atmosphere depending on the type of aerosol, height above the surface and timescale under consideration. Please, reword this sentence.

Page 1, L13: Remove directly.

Page 1, L17: Ground-based and orbital instruments have been applied...

Page 2, L4 and L5: 340 nm ... 1640 nm. Check throughout the manuscript.

Page 3, L30: Approaches to ...?

Page 4, L35: Here 'aerosol optical depth' is not capitalized (see page 2, L6).

Page 5, L4: by Dubovik and King (2000) and Nakajima et al. (1996).

Page 5, L11: I suggest deleting "As it is explained in previous section".

Page 5, L16: I suggest deleting "in detail".

Page 5, L19: I suggest deleting "really".

Page 8, L20: I suggest rewriting "huge websites".

Page 8, L26 and L31: I suggest replacing "popular" by "widely used".

Page 9, L7: I suggest deleting "really".

Page 9: I suggest replacing the title of the section "Examples" by "Performance of Caelis" (or something similar).

Page 10, L4: I suggest adding "Spain" after Madrid (as in caption of Fig. 12). Same when other locations are mentioned.

Page 11, L1-4: I suggest removing these lines.

Figure 2, caption: I suggest removing (Chen, 1976) to avoid confusion.

---

## Author Comment (AC1) · 4 Oct 2017

About the general comments:

We thank the reviewer for the constructive comments. Concerning the general comment, we want to emphasize that the work of the AERONET calibration facility at Valladolid has greatly benefited from CAELIS software. The idea to develop it was to provide tools that facilitate the management of data and instruments, and assist the calibration process, the data quality control and the network management. There was no predecessor software at Valladolid and these tasks were done manually before CAELIS. It is also important to mention that we took ideas from the management and

data control carried out in the other two AERONET calibration centers at NASA and University of Lille. We have included a sentence in the text to clarify this. We have also revised use of the English language.

About the specific comments (answers in blue):

- Page 1, L13: It is widely agreed that the overall effect of aerosols is to cool the climate. However, aerosols can also warm up the atmosphere depending on the type of aerosol, height above the surface and timescale under consideration. Please, reword this sentence.

+ Done

- Page 1, L13: Remove directly.

+ Done.

- Page 1, L17: Ground-based and orbital instruments have been applied...

+ Done.

- Page 2, L4 and L5: 340 nm ... 1640 nm. Check throughout the manuscript.

+ Done.

- Page 3, L30: Approaches to ...?

+ It refers to technical approaches and models for database design. We have clarified this in the text.

- Page 4, L35: Here 'aerosol optical depth' is not capitalized (see page 2, L6).

+ We have removed capital letters in page 2, L6, and added the acronym AOD for clarification. Now it is consistent throughout the text.

- Page 5, L4: by Dubovik and King (2000) and Nakajima et al. (1996).

+ Done.

- Page 5, L11: I suggest deleting "As it is explained in previous section".

+ Removed.

- Page 5, L16: I suggest deleting "in detail".

+ Removed.

- Page 5, L19: I suggest deleting "really".

+ Removed.

- Page 8, L20: I suggest rewriting "huge websites".

+ Rephrased to “popular websites”.

- Page 8, L26 and L31: I suggest replacing "popular" by "widely used".

+ Done.

- Page 9, L7: I suggest deleting "really".

+ Done.

- Page 9: I suggest replacing the title of the section "Examples" by "Performance of Caelis" (or something similar).

+ We have changed it to “Examples of application”.

- Page 10, L4: I suggest adding "Spain" after Madrid (as in caption of Fig. 12). Same when other locations are mentioned.

+ Done.

- Page 11, L1-4: I suggest removing these lines.

+ Suggestion accepted.

- Figure 2, caption: I suggest removing (Chen, 1976) to avoid confusion.

+ We have moved the reference to the text.

---

## Referee Comment (RC2) · Anonymous Referee #3 · 2 Nov 2017

1. General Comments:

This manuscript gives a comprehensive introduction to the software tool CAELIS, including the goal, needs analysis, software design and application examples. The work is valuable, because the manuscript not only describes how to build such a data system, but also makes us understand more details about routine operation of AERONET (such as data management, calibration and data quality control). The reviewer thinks the manuscript falls into the scope of GI. It is believed that CAELIS actually makes an important role in its facility.

However, the common issues throughout the text should be considered.

[Figure]

(1). AERONET defines data level (1.0, 1.5, 2.0) for AOD as well as skylight inversion products. CAELIS also defines the logic data levels (see Figure 3) in view of software. Are they quite different? The reviewer thinks it is wise to make them clear (not confused for readers).

(2). The parts of "Abstract" and "Summary and Conclusions" needs substantial conclusions to emphasize the benefits which CAELIS brings to the facility. For example, how long the software works, or how many instruments are under monitoring, or how quick response when malfunction happens.

2. Specific Comments:

Page 1, L2: focus on; remove "that:"

Page 1, L5: offers the scientific community a new tool

Page 1, L10:"global climate balance" means "global climate change" or "global energy balance"?

Page 2, L4: "340 nm"

Page 2, L6: "cloud-filtered" changes to "cloud screening"

Page 2, L29: "The motivation of the CAELIS . . ." is better?

Page 2, L31: ". And on the other hand" changes to "; on the other hand"

Page 3, L22: "CAELIS is composed of a database . . ."

Page 3, L30-Page 4, L12: I don't think the concept of DBMS needs many words.

Page 4, L30: The sentence "The benefit of MySQL. . ." is too complicated. Rewrite concisely and clearly.

Page 5, L26: What are "hybrid" and "cross"?

Page 6, L10: responsible for organizing . . . and deciding

Page 7, L32: CAELIS system offers users a web interface (www.caelis.uva.es) for interaction

Page 8, L20: I suggest the items such as PHP, Symfony, Javascript and JQuery and PDO can be described in shorter length.

Page 9, L27: I suggest the item "sunphotometer" or "photometer" should be internally consistent in full text.

Page 10, L20: What does "This view is highly configurable thanks to filters" mean?

Page 11, L14: AOD vs. AE

Page 12, L17: I suggest removing the sentence "This is a NASA . . ."

---

## Author Comment (AC2) · 8 Nov 2017

We thank the reviewer for the constructive comments. Concerning the general comment (1), we agree that there is possible confusion between the AERONET data levels and the CAELIS database "levels". We have therefore renamed the latter as "layers". We think it is also very descriptive of the structure of the database and can be a better term. Regarding general comment (2), we have added some sentences in the "summary and conclusions" in order to emphasize the benefits that CAELIS brings to the facility. Thanks for pointing this up.

Specific Comments (answers in blue):

[Figure]

- Page 1, L2: focus on; remove "that:"

+ Done.

- Page 1, L5: offers the scientific community a new tool

+ Done.

- Page 1, L10:"global climate balance" means "global climate change" or "global energy balance"?

+ We mean "energy balance". Corrected.

-Page 2, L4: "340 nm"

+ Done.

-Page 2, L6: "cloud-filtered" changes to "cloud screening"

+ Done.

-Page 2, L29: "The motivation of the CAELIS . . ." is better?

+ Done.

- Page 2, L31: ". And on the other hand" changes to "; on the other hand"

+ Done.

- Page 3, L22: "CAELIS is composed of a database . . ."

+ Done.

- Page 3, L30-Page 4, L12: I don't think the concept of DBMS needs many words.

+ We think this description may be necessary for those readers who come from the aerosol field. It's true that this can be obvious for readers with IT background but we prefer keeping it.

-Page 4, L30: The sentence "The benefit of MySQL. . ." is too complicated. Rewrite concisely and clearly.

+ We have rewritten the sentence.

-Page 5, L26: What are "hybrid" and "cross"?

+ These are new geometries for sky scan in Cimel instruments but the description is out of the scope here, so we just omitted them.

- Page 6, L10: responsible for organizing . . . and deciding

+ Done.

- Page 7, L32: CAELIS system offers users a web interface (www.caelis.uva.es) for Interaction

+ Rewritten as suggested.

- Page 8, L20: I suggest the items such as PHP, Symfony, Javascript and JQuery and PDO can be described in shorter length.

+ Similarly as the previous comment about DBMS, we prefer to keep this description for readers without much IT background.

- Page 9, L27: I suggest the item "sunphotometer" or "photometer" should be internally consistent in full text.

+ Thanks. We have homogenized it to "photometer", especially because latest Cimel photometers have also Moon capabilities.

- Page 10, L20: What does "This view is highly configurable thanks to filters" mean?

+ We have rewritten the sentence. We mean that the results shown in the web page can be customized by the user.

- Page 11, L14: AOD vs. AE
+ Done.

- Page 12, L17: I suggest removing the sentence "This is a NASA . . .."

+ Done.
* * *